# Micro-GLUE?: The Case for Dedicated Benchmarks in Microbiome Representation Learning

## 1 Introduction

Much recent progress in machine learning has been driven by the use of self-supervised pretraining to learn meaningful representations, especially in language modelling. These techniques have also been applied to biological data, giving rise to DNA language models Ji et al. (2021); Nguyen et al. (2023); Dalla-Torre et al. (2025); Brixi et al. (2025); Munsamy et al. (2026), protein language models Hayes et al. (2025); Lee et al. (2023) and whole-genome models Wiatrak et al. (2025); Avsec et al. (2026). An emerging frontier is the application of these ideas to microbiomes Zhang et al. (2025a); Pope et al. (2025); Medearis et al. (2026). This naturally raises the question of whether the architectures and training paradigms that succeed in natural language and molecular biology transfer effectively to communities of microbes. As this field develops, it is becoming increasingly important to evaluate which language modelling techniques are truly applicable, and to establish benchmarks that can guide progress in microbiome representation learning. In this paper, we take first steps toward evaluating representative language model architectures and the pretraining finetuning paradigm for learning meaningful microbiome representations. Our results suggest that while pretraining transfers effectively, architectural advances from language modelling may not directly translate to microbiomes, and that standard metrics such as loss, accuracy, and F1 score may be insufficient to distinguish between different learned representations. We therefore argue for the development of a dedicated microbiome benchmark to systematically assess the quality and generality of microbiome representations.

## 2 Results and Discussion

We compared transformer based models spanning major language modeling paradigms: GPT-2, RoBERTa, and Llama, matched to approximately 9M parameters with the same width and depth, consistent with a prior GPT-2 based microbiome model trained on the same dataset Zhang et al. (2025b). Because Llama architectures with identical width and depth to the other models contain more trainable parameters, we included two Llama variants: Llama-9M, matched by parameter count, and Llama-13M, matched by width and depth. Following the approach in Zhang et al. (2025b), microbiomes were encoded as a sequence of genera ordered by relative abundance, enabling treatment analogous to token sequences in language modeling.

All models were pretrained on a large microbiome corpus, Microcorpus-260K [Zhang et al. (2025b); Richardson et al. (2023)] using either next-token prediction (GPT-2, Llama) or masked language modeling (RoBERTa), with a 90%/10% train/validate split. Mean pooling over token embeddings from the last hidden layer of each model was used to generate sequence embeddings for the validation set. Dimensionality reduction demonstrates that these learned microbiome representations are meaningful (Figures 1, 2), with visible differences in the embeddings learned by the different model architectures. We then fine-tuned on a supervised biome classification task via multi-head classification of five MGnify biome labels Richardson et al. (2023) using the 26k samples held out from pretraining, again with a 90%/10% train/validate split. Performance is reported as mean accuracy and F1 score across biome heads on the validation set (Table 1). For all architectures, pretrained models substantially outperform randomly initialized baselines, demonstrating that self-supervised learning yields meaningful representations for microbiome data. We also evaluated a setting in which transformer layers were frozen and only the task-specific classification heads were trained. Here, pretrained models retained a clear advantage over non-pretrained baselines, further demonstrating that pretrained representations encode biologically relevant structure. In contrast, non-pretrained frozen models performed poorly, confirming that the capacity of the classification

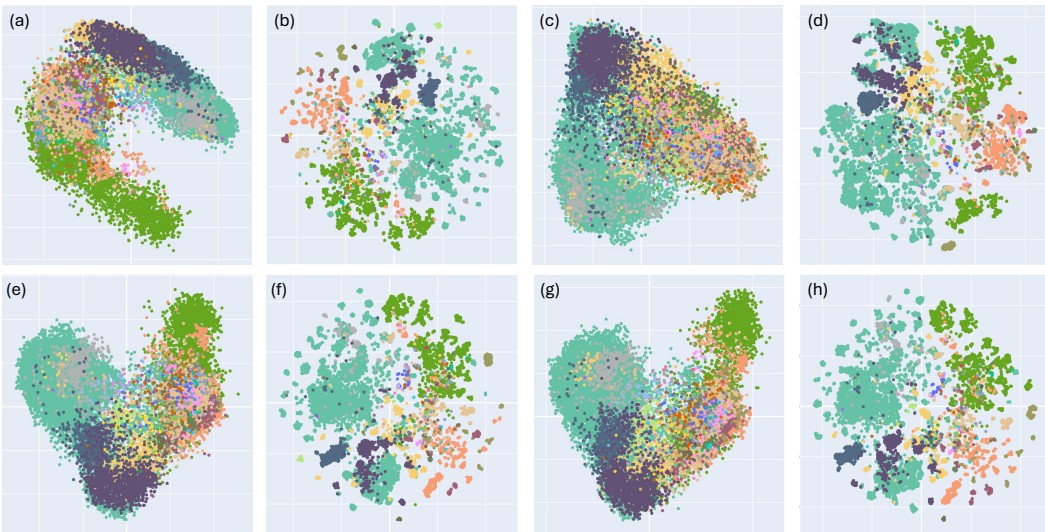

Figure 1: Dimensionality reduction of validation set embeddings from pretrained models coloured by MGnify biome label. (a,b) GPT2-9M, (c,d) RoBERTa, (e,f) Llama-9M, and (g,h) Llama-13M. PCA and t-SNE projections respectively.

| Model | Config | Loss | Acc. Mean | F1 Mean |
|---|---|---|---|---|
| GPT2-9m | non-pretrained, frozen | 0.524 | 0.843 | 0.520 |
| | pretrained, frozen | **0.174** | **0.953** | **0.811** |
| | non-pretrained, non-frozen | 0.249 | 0.932 | 0.690 |
| | pretrained, non-frozen | **0.178** | **0.953** | **0.762** |
| Llama-9m | non-pretrained, frozen | 0.468 | 0.871 | 0.600 |
| | pretrained, frozen | **0.215** | **0.941** | **0.758** |
| | non-pretrained, non-frozen | 0.225 | 0.943 | 0.714 |
| | pretrained, non-frozen | **0.167** | **0.952** | **0.775** |
| Llama-13m | non-pretrained, frozen | 0.483 | 0.862 | 0.578 |
| | pretrained, frozen | **0.220** | **0.939** | **0.748** |
| | non-pretrained, non-frozen | 0.240 | 0.934 | 0.676 |
| | pretrained, non-frozen | **0.159** | **0.956** | **0.783** |
| RoBERTa-9m | non-pretrained, frozen | 0.556 | 0.832 | 0.413 |
| | pretrained, frozen | **0.263** | **0.927** | **0.741** |
| | non-pretrained, non-frozen | 0.319 | 0.911 | 0.583 |
| | pretrained, non-frozen | **0.175** | **0.957** | **0.742** |

Table 1: Validation results across models and training configurations.

heads alone are insufficient to perform the task. However, we observe no substantial performance differences between architectures. Most interestingly, we don't see any improvement from using the more modern Llama architecture over GPT2.

Taken together, we have demonstrated evidence that the pretraining-finetuning paradigm can be applied to learning microbiome representations, but that architectural advances from language modelling may not directly translate. These findings point to a limitation of standard downstream evaluation protocols in microbiome representation learning: commonly used accuracy and F1 based metrics fail to distinguish between model architectures. As seen in Figure 1, we do not claim that these architectures necessarily learn identical representations, but rather that current evaluation protocols are insufficient to determine when and how learned microbiome representations differ. This motivates the need for benchmarks that more directly probe representation quality and biological meaning. Frozen transformer transfer emerges as a simple, but potentially informative, probe of representation quality that could be incorporated into future microbiome benchmarks. Establishing standardized datasets, splits, and evaluation protocols tailored to microbiome data would provide a foundation for systematic progress and more interpretable comparisons, analogous to the role played by GLUE or BIG-bench in natural language processing.

MEANINGFULNESS STATEMENT

A meaningful biological representation should capture underlying structure, diversity, and functional relationships, not just enable prediction. Microbiome data captures complex communities that directly influence host health, disease, and environmental processes yet remain comparatively underexplored as a target for foundation models, and lack standardized benchmarks for evaluating learned representations. This makes it difficult to assess whether advances in model architecture meaningfully translate to microbiome understanding. Our work shows that while self-supervised pretraining yields meaningful microbiome representations, standard evaluations fail to distinguish architectural differences. We therefore motivate microbiome specific benchmarks that better assess biological meaning, robustness, and generalization in microbiome representations.

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

## A  APPENDIX

- Marine
- Soil
- Digestive system
- Skin
- Continuous culture
- Gastrointestinal tract
- Rhizosphere
- Root
- Water and sludge
- Freshwater
- Sediment
- Dairy products
- Reproductive system
- Respiratory system
- Phylloplane

- Simulated communities (microbial mixture)
- Persistent organic pollutants (POP)
- Estuary
- Rhizoplane
- Industrial wastewater
- Fermented beverages
- Defined media
- Deep subsurface
- Activated Sludge
- Non-marine Saline and Alkaline
- Thermal springs
- Simulated communities (sequence read mixture)
- Cnidaria
- Fermented vegetables
- Circulatory system

- Landfill
- Lentic
- Brown Algae
- Fermented seafood
- Composting
- Oral cavity
- Echinodermata
- Wet fermentation
- Nutrient removal
- Geologic
- Terephthalate
- Aquaculture
- Oil reservoir

Figure 2: Biome labels key

