# OpenReview forum: "MICRO-GLUE?: THE CASE FOR DEDICATED BENCHMARKS IN MICROBIOME REPRESENTATION LEARNING"
_ICLR.cc/2026/Workshop/LMRL — Submitted to ICLR 2026 Workshop LMRL_

### Official Review · Reviewer_TkFi · 2026-02-20
**MICRO-GLUE? THE CASE FOR DEDICATED BENCHMARKS IN MICROBIOME REPRESENTATION LEARNING**

**Rating:** 4
**Confidence:** 4

**Review:**

# Review: MICRO-GLUE? THE CASE FOR DEDICATED BENCHMARKS IN MICROBIOME REPRESENTATION LEARNING

## Summary

This paper identifies the lack of standardised benchmarks for microbiome foundation models and argues for a dedicated evaluation framework analogous to GLUE in NLP. To motivate this, the authors compare three transformer architectures (GPT-2, RoBERTa, Llama) matched at ~9M parameters, pretrained on Microcorpus-260K, and fine-tuned on biome classification. They show that pretraining consistently helps but that standard metrics (accuracy, F1) cannot distinguish between architectures, concluding that more targeted benchmarks are needed.

## Strong Points

- **S1. Timely problem.** The absence of standardised benchmarks for microbiome representation learning is a genuine gap, and the paper articulates this clearly as more models emerge in this space (MGM, biomeGPT, MGP).
- **S2. Controlled setup.** Architectures are matched at ~9M parameters, with an additional Llama variant at 13M to control for width/depth differences. The four-way comparison (pretrained/non-pretrained × frozen/non-frozen) is well-structured.
- **S3. Clear writing.** The paper is concise, claims are plainly stated, and the experimental setup is reproducible.

## Weak Points

- **W1. Experimental design does not test the central claim (Major).** GPT-2, RoBERTa, and Llama share the same core mechanism and differ only in secondary design choices (positional encoding, activation function, attention grouping). At 9M parameters these differences are marginal. The design cannot distinguish between: (a) benchmarks are inadequate, or (b) the compared models are functionally equivalent at this scale.
- **W2. No existing microbiome models are benchmarked (Major).** The paper cites MGM, MGP, and biomeGPT as motivation, altough similar approaches are evaluated none of the cited models are benchmarked. The claim that current evaluation fails to differentiate models in the field should be tested on models in the field.
- **W3. No non-deep learning baselines (Major).**  Classical methods (RF, XGBoost) are known to be competitive on such data and are used in cited works. Without them, it is unclear whether the classification task is too easy to be discriminative or whether learned representations add value over raw features.
- **W4. Qualitative-only embedding analysis (Moderate).** PCA/t-SNE plots are the sole evidence that representations are "meaningful." No quantitative metrics (silhouette score, k-NN purity) are provided. Dimensionality reduction of high-dimensional embeddings onto coarse labels will almost always show visual separation.
- **W5. Frozen random baseline is an artificially low bar (Moderate).** A frozen randomly initialised transformer applies an arbitrary nonlinear transformation that can destroy input structure. Outperforming it shows pretraining encodes *something*, but this is a weaker claim than the paper implies.

## Recommendation

**Reject.**

1. The central argument — that better benchmarks are needed — is supported only by the observation that three closely related transformer variants perform similarly. This is expected and does not implicate the evaluation protocol.
2. The experiments are disconnected from the paper's own motivation: no existing microbiome models and no classical baselines are evaluated.

**Supporting arguments:**
- The paper identifies a valid problem but does not provide evidence that the problem exists for the reason it claims. An alternative and more parsimonious explanation is that the architectures are near-identical at this scale.
- The frozen non-pretrained baseline inflates the apparent benefit of pretraining by comparing against a setting where the task may be harder than learning from the raw data. A simple classical baseline would contextualise all results.
- The paper calls for a Micro-GLUE but proposes no concrete tasks, datasets, or metrics beyond the single biome classification task already used.

## Questions for Authors

1. How do non-deep learning baselines (e.g., RF or XGBoost on genus abundance vectors) perform on the biome classification task?
2. What evidence would rule out that GPT-2, RoBERTa, and Llama at 9M parameters are simply too similar to produce meaningfully different representations, regardless of evaluation protocol?
3. Have you considered including existing microbiome foundation models (MGM, biomeGPT) in the comparison?
4. How does a classification head trained directly on raw abundance features compare to the frozen pretrained representations?

## Additional Feedback

These suggestions are intended to help improve the work and are not part of the decision assessment:

- A single classical ML baseline would immediately ground all results and strengthen the motivation for dedicated benchmarks.
- Including one or two existing microbiome models would directly demonstrate the evaluation gap the paper identifies.
- Proposing even a preliminary set of tasks beyond biome classification would make the Micro-GLUE contribution concrete rather than aspirational.

## Confidence

4 — High: I am confident in my assessment, but not absolutely certain.

## Code of Ethics

No potential violation identified.

---

### Official Review · Reviewer_zjpE · 2026-02-24
**Promising Direction; A Review of Micro-GLUE**

**Rating:** 5
**Confidence:** 3

**Review:**

**Minor Comments:**

- Zhang et al. (2025) is cited twice - reconcile into a single citation.

**Major Comments:**

1. **Task is never clearly defined.** What does the input data actually look like? How is a microbiome encoded as a sequence, and what do the five biome labels represent biologically? A figure or concrete example illustrating the input-output structure is needed.

2. **Obvious baseline is missing.** Microcorpus-260K comes from Zhang et al. (2025), who trained a microbiome foundation model on this exact data. Not comparing against it directly is a significant omission, and raises the question of whether this work would be better positioned as a benchmarking component of that paper rather than a standalone contribution.

3. **'Vision' is too vague.** The finding that architectural differences are indistinguishable under current metrics is interesting, but the paper doesn't tell the reader what to do next. What tasks, splits, and evaluation criteria would a "Micro-GLUE" actually include? Even a preliminary framework would substantially strengthen the contribution.

4. **Metabolic modeling literature is missing.** A large body of work on microbial community modeling (flux balance analysis, community metabolic modeling) has developed principled ways of thinking about microbial function and ecological roles. This could directly inform what a meaningful microbiome representation should capture, and suggest richer evaluation tasks beyond biome classification.

---

### Meta-Review · Area_Chair_sBeX · 2026-02-28

**Recommendation:** Reject
**Confidence:** 3

**Metareview:**

While I generally like papers that argue for better benchmarks, the concerns raised by the reviewers are correct - in particular, "The central argument — that better benchmarks are needed — is supported only by the observation that three closely related transformer variants perform similarly. This is expected and does not implicate the evaluation protocol." from TkFi is the key point.

---

### Decision · Program_Chairs · 2026-03-02

**Decision:**

Reject

**Comment:**

Please see the meta-review.